# Machine-Learning Applications in Energy Efficiency: A Bibliometric Approach and Research Agenda

**Alejandro Valencia-Arias** [1,*], **Vanessa García-Pineda** [2], **Juan David González-Ruiz** [3], **Carlos Javier Medina-Valderrama** [4] and **Raúl Bao García** [5]

1 Escuela de Ingeniería Industrial, Universidad Señor de Sipán, Chiclayo 14001, Peru
2 Facultad de Ingenierías, Instituto Tecnológico Metropolitano, Medellín 050004, Colombia; vanessagarciap@itm.edu.co
3 Departamento de Economía, Universidad Nacional de Colombia, Medellín 050004, Colombia; jdgonza3@unal.edu.co
4 Department of General Studies, Universidad Señor de Sipán, Chiclayo 14001, Peru
5 Academic Vice Rectory, Universidad de San Martín de Porres, Lima 15011, Peru
* Correspondence: valenciajho@crece.uss.edu.pe; Tel.: +51-3002567977

**Abstract:** The high demand for energy resources due to the increasing number of electronic devices has prompted the constant search for different or alternative energy sources to reduce energy consumption, aiming to meet the high demand for energy without exceeding the consumption of natural sources. In this context, the objective of this study was to examine research trends in the machine-learning-based design of electrical and electronic devices. The methodological approach was based on the analysis of 152 academic documents on this topic selected from Scopus and Web of Science in accordance with the preferred reporting items for systematic reviews and meta-analyses (PRISMA) statement. Quantity, quality, and structural indicators were calculated to contextualize its thematic evolution. The results showed a growing interest in the subject since 2019, mainly in the United States and China, which stand out as world powers in the information and communication technology industry. Moreover, most studies focused on developing devices for controlling, monitoring and reducing energy consumption, mainly in 5G and thermal comfort devices, primarily using deep-learning techniques.

**Keywords:** energy efficiency; energy consumption; sustainability; PRISMA; machine learning; deep learning

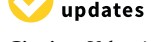



## 1. Introduction

Environmental deterioration has been increasing due to global economic and industrial development, among other factors. The increase in fossil fuel production has had a considerable impact on health risks and on the limited resources of the planet [1]. Yet, despite the ongoing generation of different energy transformation processes, in the immediate future, fossil fuels will continue to play a key role in the market [2]. Consequently, the Organisation for Economic Co-operation and Development (OECD) and the Sustainable Development Goals proposed by the United Nations (UN) have stated that countries must urgently seek alternatives to promote the use of less expensive, cleaner, and more affordable energy sources [3,4]. In this sense, we consider that energy transition is a fundamental factor for achieving progress as a smart territory and contributes to the construction of the city, bearing in mind that from the United Nation's Agenda, one of the Sustainable Development Goals (SDGs), specifically SDG 7, raises the need for the generation of affordable and non-polluting energy, while SDG 11 raises for the construction of sustainable city and communities. Thus, given that by creating methods for generating and optimizing efficient energy resources through technological tools, it is possible to contribute to the development of new forms of non-polluting energy generation and access to the entire society.

The urgent need for energy alternatives primarily results from environmental damage. This environmental damage raises concern regarding the progress of modern cities because prevention measures must be applied to various economic sectors [5]. Society is currently demanding increased fuel production and an economy without adverse environmental impacts [6], thus requiring technological advances and investments in modern automation for designing devices with low carbon emissions, increasing infrastructure availability and fostering energy-efficient services [7]. Energy efficiency is a metric developed to balance performance and equity towards reducing consumption and mitigating the impact of inappropriately using energy resources, thereby achieving sustainability in modern society [8].

For this purpose, devices and technologies must be designed to facilitate efficient energy consumption in the context of smart cities, considering their benefits in industry, transport, and urban infrastructure management. Buildings are the largest energy consumers [4,9], as they comprise most urban infrastructures and are responsible for a large amount of carbon emissions [7]. In fact, the construction sector accounts for almost a third of the world's final energy consumption [10]. Therefore, recent research efforts have focused on addressing issues related to smart cities and the Internet of Things (IoT), such as environmental conservation [11], because the habits of residents can disturb energy performance.

In line with the above, technological progress has enabled the development of techniques for prolonging the use of devices while maintaining moderate energy resource consumption [12]. Accordingly, recent advances in both hardware and software have led to the development of smart devices, making it possible to build large, sustainable infrastructures [13]. In turn, advances in Industry 4.0 through the development of artificial intelligence (AI) has led to major advances in different fields and sectors, including the energy sector [14].

As an AI branch, machine learning (ML) has various applications in smart cities and buildings, which use IoT-related Fourth Industrial Revolution technologies to connect and automate various daily functions. Therefore, these technologies are crucial for integrating large systems that contribute to safety and maintenance through their control using mobile devices and computers [15].

Current industrial performance promotes data generation in large quantities. These data are mostly overlooked, and recent developments in ML within the big data environment remain underexploited [9,16]. However, ML-based models are being increasingly used to predict patterns in occupancy routines and behaviors towards enhancing energy efficiency [17]. These models have aroused interest in applying ML in integrated systems to produce a faster response and reduce network load [18], thus opening up new opportunities for creating intelligent and efficient manufacturing systems [19].

Various industries have recently sought to reduce the consumption of different types of fuel. For this purpose, recent research advances have brought about developments in intelligent management systems for energy efficiency by collecting data on energy consumption [20]. Notwithstanding the high contribution of buildings to global energy consumption and greenhouse gas emissions, projecting consumption patterns is beneficial for the industry [21]. However, the industry is not the only sector that can benefit from ML technologies. For example, ML can be applied in the energy management of ships and various other means of transport, helping to increase profits and mitigate environmental impacts by reducing $CO_2$ emissions caused by transportation activities [22]. Cellular networks have also revolutionised daily life, and fifth generation (5G) mobile networks will enable a high degree of automation. This automation will provide relevant data for managing energy consumption per capita through tools that contribute to sustainability goals of the information and communications technology (ICT) sector [23].

The mass use of IoT devices implies a considerable network expansion and, hence, the use of a high number of sensors for gathering and sending data. These sensors require a constant power supply for their operation [24]. Accordingly, studies have assessed

the development of different ML models to reduce IoT energy consumption, starting a discussion on the effectiveness of predicting energy consumption and on the useful life of these devices in the network [25]. With the paradigm and rapid advancement of IoT, constant data exchange requires energy consumption, which is limited in devices such as sensors. For this reason, recent research has focused on searching for ML models and bio-inspired algorithms to reduce energy consumption in IoT [26]. ML methods have also enabled device manufacturers to use these data to carefully calculate power consumption from the design stage [27].

Implementation costs, in turn, affect the decisions of manufacturing companies to invest in energy-efficient technologies, as long as they provide effective production costs and minimise carbon dioxide emissions and the need for energy imports [4]. Therefore, ML approaches to the evaluation of industrial energy use and cost savings can affect the decisions of manufacturers to implement energy efficiency improvements in the industry [28].

The findings discussed above account for the growing interest in the search for solutions to environmental problems, using technological advances of the Fourth Industrial Revolution. In this context, ML technologies play a key role in developing energy-efficient alternatives. For this reason, the aim of this study is to conduct a bibliometric analysis of the literature to identify research trends regarding the development of ML applications in energy efficiency. The incessant increase in information and knowledge on the subject highlights the scientific and technological need to evaluate studies on this subject [29] by addressing metrics associated with research quantity and quality by author, country, journal, and emerging concepts. These parameters will be key to the future development of this subject. To achieve the aim of this study, the following research questions are posed:

RQ1: How has the scientific literature on the use of ML for energy efficiency evolved historically?
RQ2: What are the main research references for the use of ML for energy efficiency?
RQ3: What is the thematic evolution of the scientific production on the use of ML for energy efficiency?
RQ4: What are the main thematic clusters of the use of ML for energy efficiency?
RQ5: What are the established and emerging keywords in the research field of the use of ML for energy efficiency?
RQ6: What are the relevant topics for the design of a research agenda for the use of ML for energy efficiency?

The literature review process is described below in the Materials and Methods section. Subsequently, the research results are presented and discussed in the corresponding sections, thereby proposing a research agenda. Last, the main findings and contributions pertaining to the aims of this study are outlined in the Conclusions.

## 2. Materials and Methods

To address the research aims and questions outlined above, this exploratory study employed a bibliometric analysis, because this method facilitates the evaluation of the research output. For this purpose, scientific metadata, including title, authors, journals, affiliations, and keywords, were analysed using bibliometric indicators, such as the quantity (measuring scientific productivity), quality (assessing the impact of publications based on citation counts), or structural (highlighting associations between authors and keyword relationships) indicators [30]. To ensure a detailed and replicable method, the preferred reporting items for systematic reviews and meta-analysis (PRISMA) statements, updated in 2020 [31], were followed, as described below.

### 2.1. Eligibility Criteria

According to the PRISMA 2020 statement, the eligibility criteria defined the inclusion criteria and the exclusion criteria [31]. Accordingly, the inclusion criteria for this study regarding the use of ML for energy efficiency were articles that included the combination

of energy efficiency and ML in their title, keywords, and main scientific metadata to ensure that the records retrieved from the databases addressed the topic under study.

Conversely, the exclusion criteria were implemented in three consecutive screening phases. In the first phase, all articles identified due to typographical errors in the indexing of each database were excluded from the sample. Subsequently, records without full-text access were also excluded. However, this process is applicable to systematic literature reviews as rigorous methods of analysis of the complete content of research articles for specific purposes; in contrast, bibliometric analysis examined the metadata of scientific activity [30]. Last, in the third phase of screening, all incompletely indexed documents were excluded from the sample because they limited the bibliometric analysis. Depending on the type of document, documents that did not comply with rigorous methodological processes and thus did not address the research aims were also excluded from the sample.

*2.2. Sources of Information*

To obtain scientific documents on the use of ML for energy efficiency, two of the main academic and scientific databases, namely Scopus and Web of Science, were selected for their thematic coverage, easy interface, rigorous evaluation and indexing processes, and metadata registration [32].

*2.3. Search Strategy*

In the search for articles in the selected databases, the PRISMA 2020 statement highlights the importance of designing a pertinent search strategy considering the search characteristics of each source of information and, above all, the inclusion criteria [31]. Accordingly, records were extracted using the following search equations:

Scopus database search: (TITLE ("energy efficiency")) AND (TITLE ("machine learning")) OR (KEY ("energy efficiency")) AND (KEY ("machine learning")); and
Web of Science database search: (TI = ("energy efficiency")) AND (TI = ("machine learning")) OR (AK = ("energy efficiency")) AND (AK = ("machine learning")).

*2.4. Data Management*

The search strategies were implemented in each database, yielding a total of 234 records on the use of ML for energy efficiency. The data were stored in a Microsoft Excel® spreadsheet. Using this tool, the exclusion criteria were applied, selecting a total of 152 articles. Lastly, using the open-source software VOSviewer, the bibliometric (quantity, quality and structural) indicators detailed above were applied to analyse the included articles.

*2.5. Selection Process*

To reduce bias in the inclusion and exclusion of articles and in accordance with the PRISMA 2020 statement [31], each author independently performed the searches as well as the data extraction and storage process. Additionally, the three screening phases were completed independently. The resulting differences were analysed together by all of the authors until reaching a consensus. This methodological design is summarised in the flow diagram shown in Figure 1.

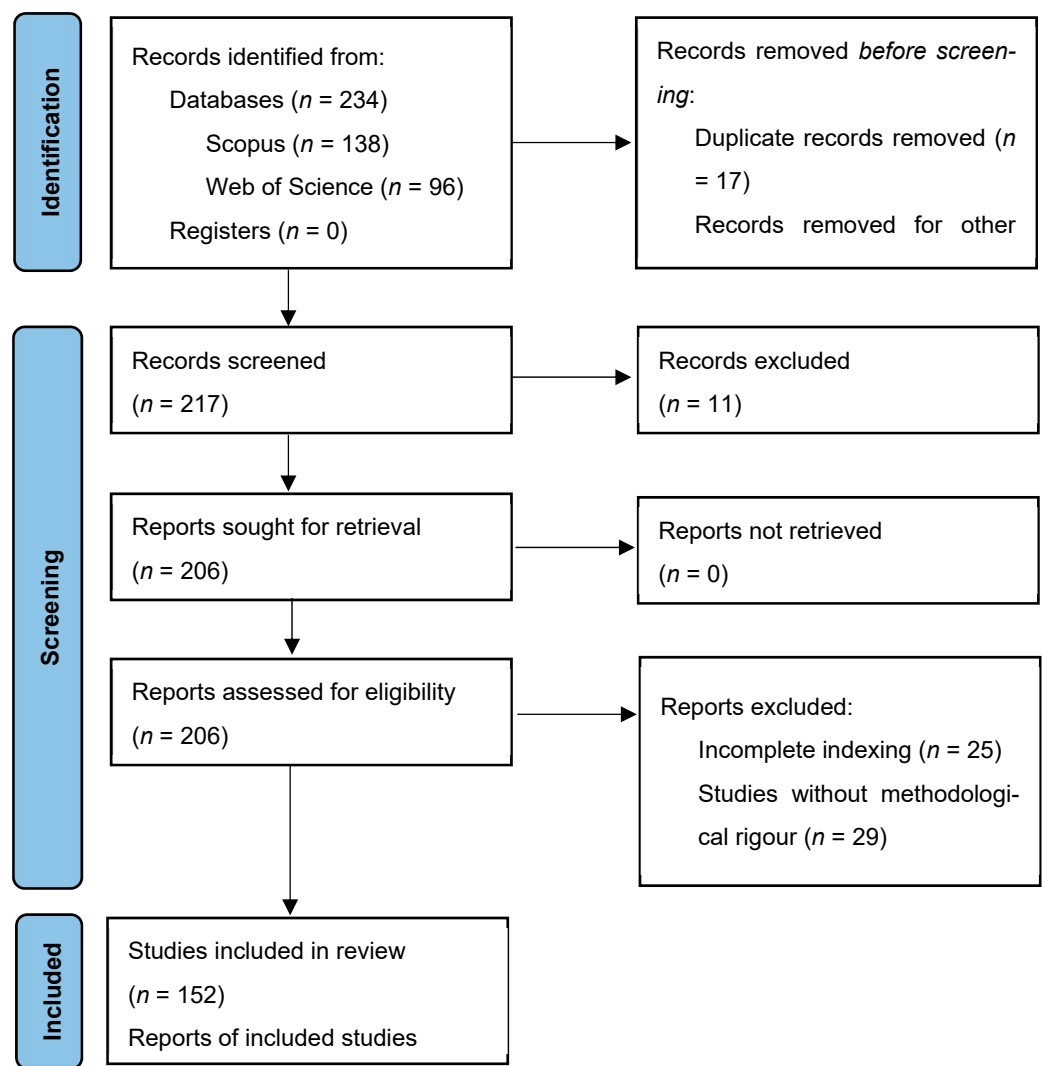

**Figure 1.** PRISMA-2020 methodological design. Source: the authors, based on data retrieved from Scopus and Web of Science.

The first stage (identification) consisted of searches for and the identification of articles in both databases, in addition to the elimination of duplicate records. Subsequently, the screening stage consisted of three consecutive phases of exclusion. A total of 152 articles were included in the final sample for bibliometric analysis.

## 3. Results

As shown in Figure 2, the indicators highlight increasing interest in the use of ML for energy efficiency from 2019 to 2022. The year 2022 had the highest number of publications on this topic, totalling 43 articles on the use of ML to predict usage patterns and behaviours, particularly focusing on applications and benefits to building systems resulting from improvements in the energy efficiency of utilities in public spaces, in indoor air quality, and in thermal comfort [17].

## Publications per year

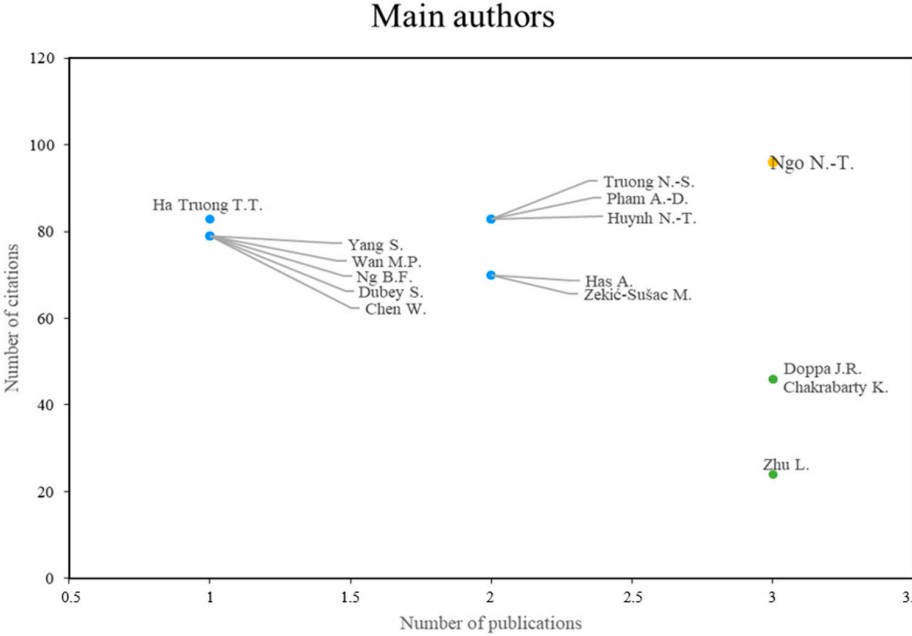

**Figure 2.** Publications per year. Source: the authors, based on data from Scopus.

In 2021, the year with the second-most publications (*n* = 40), the studies primarily examined energy efficiency heterogeneity in China via data encapsulation analysis (DEA) and ML, showing the indirect effects of the regional accumulation of low-carbon technological innovation for energy efficiency using the dynamic spatial Durbin model (DSDM) [33]. In addition, in this year, publications also focused on topics such as green computing, for which different systems were investigated to generate energy-efficient 5G networks and network infrastructures [34]. The energy efficiency of Edge-to-Cloud was also investigated using ML [35], in addition to research aimed at optimising energy efficiency driven by quantum technology for next-generation communication systems [36].

In 2020 and 2019, studies focused on the use of ML for the energy efficiency of urban infrastructure and for urban and industrial development [11,37–40].

Subsequently, the main authors of scientific studies on the use of ML in energy efficiency were analysed by comparing the number of publications, associated with scientific productivity, and the number of citations, assessing the impact on the scientific community (Figure 3).

**Figure 3.** Publications by author. Source: the authors, based on data from Scopus.

As shown in Figure 3, Ngo N-T is currently the most prolific author on the subject, with three studies and a total of 96 citations. His publications include a framework for intelligent spatial decision support systems (SDSS) integrating big data analytics from smart grids and cloud computing to generate energy efficiency based on a layered architecture that includes smart grids and data collection, warehouse analytics, and web portals [41].

The most cited article co-authored by Ngo N-T used ML to predict energy consumption in multiple buildings to improve energy efficiency and sustainability. Using the random forest (RF) model, this study contributes to the state of the art by testing the generalisation and effectiveness of ML models in predicting energy consumption patterns in a building [21].

Ngo N-T is followed by a group of authors with high total citation counts and thus they are deemed high-impact authors in the scientific community [42], although they are not among the most productive or prolific. This group includes authors such as Pham A-D, Truong N-S, and Huynh N-T, with a combined total of 83 citations. Their research focused on proposing an RF-based predictive model to accurately predict short-term energy consumption per hour in many buildings [21].

The last group includes authors who are, conversely, considered benchmarks of academic productivity, thanks to their total number of publications on the use of ML for energy efficiency, despite lacking a high citation count. Among them, Doppa JR and Chakrabarty K have published three articles on this topic, most notably a study exploring a small-world (SW) network-based three-dimensional (3D) Network-on-Chip (NoC) architecture leveraging ML to intelligently explore the design space to optimise the placement of planar and vertical communication links to save energy, demonstrating that optimised 3D SW NoC designs considerably outperform their 3D mesh counterparts [42]. Another author who also stands out for high scientific productivity is L. Zhu, who has published three studies mainly focused on energy efficiency in wireless networks by applying deep reinforcement learning (DRL) techniques [43].

This bibliometric analysis identified the most popular scientific journals on the use of ML for energy efficiency and compared the number of publications and citations as indicators of research productivity and impact, respectively (Figure 4).

**Figure 4.** Publications by journal. Source: the authors, based on data from Scopus.

The main scientific journal is *Applied Energy*, which accounts for a total of four publications and has been ranked among the most productive in this field of research. These four studies have amassed more than 130 citations; therefore, *Applied Energy* stands out as the journal with the highest impact in the scientific community regarding the use of ML for energy efficiency. Among the studies published in this journal, a seminal study experimentally modelled the financial returns on building energy retrofit investments in more than 3600 multifamily and commercial buildings in New York City, using a comprehensive database of energy audits and renovation work extracted from city records using a natural language processing (NLP) algorithm [44]. The second most cited publication analysed the energy efficiency of an air extraction system using a supervised ML model [45].

Another key study published in Applied Energy addressed newer or topical components [46]. The authors of that study proposed a predictive control system based on an adaptive ML model for building automation and control applications.

The second most important journal in this field of research is Energies, which accounts for a total of 90 citations. As such, Energies also ranks second in terms of impact in this academic and scientific community. With a total of 10 publications, Energies has the highest scientific productivity, i.e., the journal that disseminates the most knowledge about the use of ML for energy efficiency. A recent publication focused on increasing the efficiency and reliability of protocols in current applications by redesigning two well-known energy efficiency protocols, namely low-energy adaptive clustering hierarchy (LEACH) and variable energy-efficiency sensor routing (EESR), considering neural networks [47].

Another key study published in Energies [48] provided energy efficiency solutions for buildings consisting of automated fault diagnosis for various energy consumption components, thus promoting early detection and energy efficiency optimisation. For this purpose, the study proposed a hybrid generative adversarial network (GAN) that combined Wasserstein GAN with traditional classifiers to perform fault diagnoses, mimicking real-world scenarios with limited faulty training samples in the training process [48].

More recently, despite not ranking among the journals with the highest impact or the highest total number of citations, a group of journals has stood out for publishing several papers on ML and energy efficiency, e.g., Energy Efficiency, which has published four articles on the subject, of which the two most cited articles have broadened the knowledge on the use of data with ML techniques for home energy efficiency management [46,49].

Another journal that stands out in this group of scientifically productive journals is Energy and Buildings, whose most cited article aimed to predict occupant workstation level energy use in commercial buildings to increase the energy efficiency of buildings and to promote their automation [50]. The second most cited publication in this journal investigated the data-driven optimisation of building layouts for energy efficiency [51].

This bibliometric analysis evaluates the main countries that generate knowledge on the use of ML for energy efficiency and that stand out as benchmarks in terms of schools of thought by comparing the number of publications and citations (Figure 5).

As shown in Figure 5, the leading country in generating high-impact knowledge on the use of ML for energy efficiency is the United States, with a total of 37 articles and 384 citations. As such, the United States is the most productive and the most cited country in the field, with the strongest impact on this scientific community. A seminal study in the United States conducted a comprehensive and detailed systematic review of AI-based techniques used to build control systems and to evaluate the results and implementation of these techniques in peer-reviewed publications [52]. Another key study published in recent years focused on presenting methods for relating lighting zone energy to zone-level occupant dynamics, simulating energy consumption of a lighting system based on this relationship, and optimising the layouts of buildings [51].

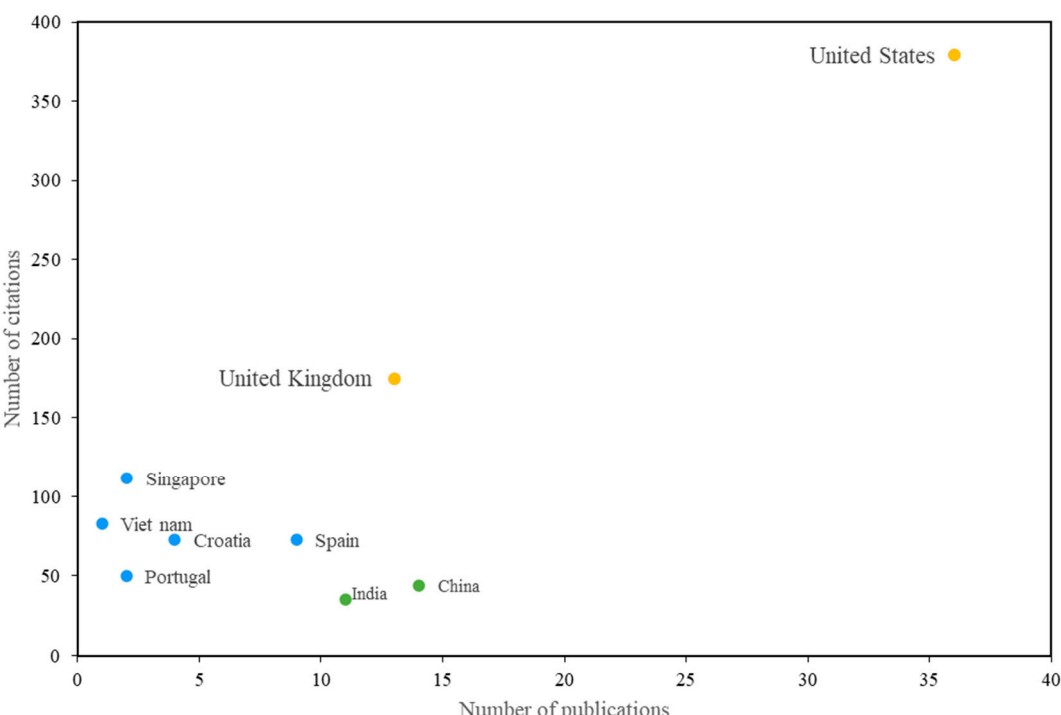

**Figure 5.** Publications by country. Source: the authors, based on data from Scopus.

The second leading country in research output on the subject is the United Kingdom. With 13 publications, the United Kingdom ranks among the most productive countries and is placed second among the countries with the strongest scientific impact on this field, with a total of 160 citations. The leading publication from this country provides an overview of AI-based energy efficiency systems for domestic application and explains how micro-moments can provide an accurate understanding of user behaviour and lead to more effective recommendations [53].

Lastly, there is a group of countries that may not rank among the most cited in this scientific community, but who account for a high number of publications and are thus benchmarks in terms of productivity and the generation of knowledge on the use of ML for energy efficiency. In this group of countries, China, with 14 articles and the second most productive country behind the United States, stands out. Its most representative studies have presented a new convolutional computing paradigm based on the non-volatile storage (NOR) flash array (NFA), which can execute two-dimensional (2D) convolution in a single clock cycle [1], and proposed an innovative way of using liquefied natural gas (LNG) cold energy that can solve these three problems simultaneously [54].

## 4. Discussion

The network of authors is mainly connected by collaborations between X. Liu, L. Liu, Y. Liu, and C. Liu (Figure 6). Xiaoyan Liu has published since 1995, primarily focusing her research on signal processing optimisation in computing devices. One of her studies on energy efficiency used convolution computing to increase signal speed and to improve processing energy efficiency [55]. Another interesting collaboration study from this group of authors was published by Y. Liu, who proposed using deep learning (DL) techniques to select microwaves so as to enhance energy efficiency for cellular-assisted vehicular networks [56].

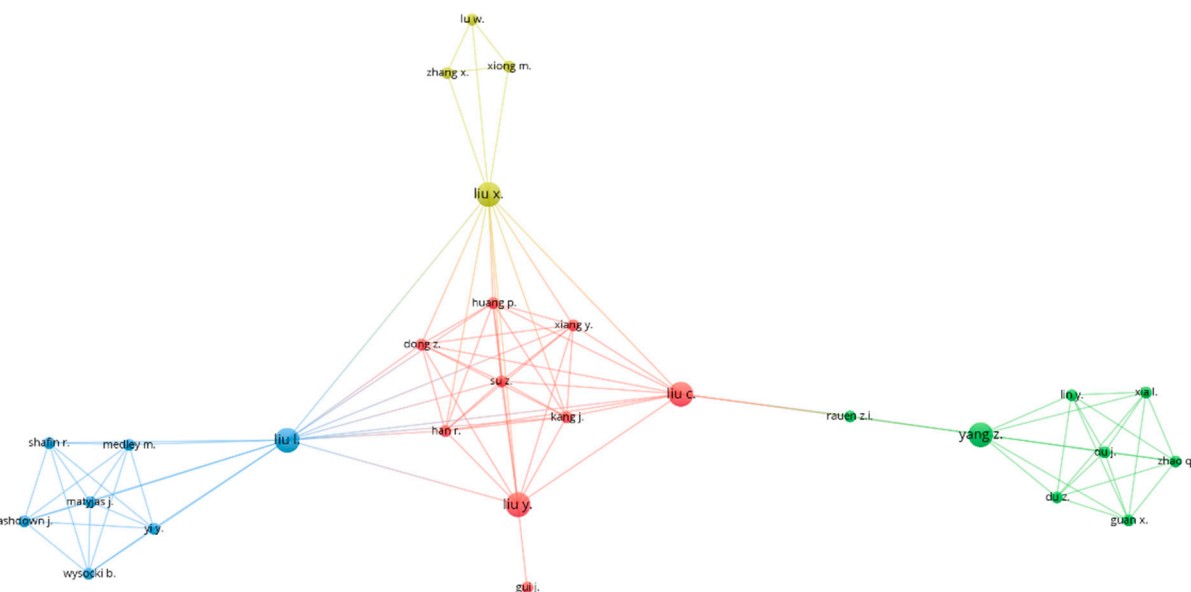

**Figure 6.** Collaborations between authors. Source: the authors, based on data from Scopus.

Another interesting study developed an energy efficient spectrum-sensing method for multiple-input multiple-output and orthogonal frequency division multiplexing (MIMO-OFDM) systems to take advantage of the spatial and temporal correlations of the environment. The aim of this approach was to address the problematic lack of data when applying ML techniques to MIMO-OFDM systems [57]. A highly relevant study and the most cited in Scopus (175) was authored by C. Liu and colleagues, who reported a deep reinforcement learning (DRL) method for developing smart mobile terminals for industrial IoT applications to facilitate data collection and sharing with minimal energy consumption [58]. In their proposals, these authors considered not only the energy efficiency of such applications and the use of AI techniques, but also modulation techniques, most of which focused on telecommunications applications or on the use of information technology (IT) devices in infrastructure.

As shown in the keyword network (Figure 7), the most commonly used term in most studies is energy consumption and its association with DL. For example, in research conducted by [59], the authors implemented an energy-efficient carbon fibre manufacturing system through waste heat recovery using data on energy consumption factors to model the total energy and its balance in the thermal stabilisation step to subsequently develop a model based on DL and neural networks for predicting energy consumption. Another study on energy consumption, albeit with the term 5G networks, analysed the energy efficiency of 5G networks, considering factors such as massive MIMO, operator design, sleep modes, and the use of ML techniques [23].

DL is one of the most commonly used terms in this field of research because various studies report using data from DRL techniques to optimise cooling systems [60], building energy efficiency [10], and energy consumption management in smart homes [49] and, from DL approaches, to develop energy efficiency models [61]. Accordingly, some studies have applied DL techniques in smart buildings and cities to develop energy efficiency management systems focused on smart [9] and sustainable [62] cities.

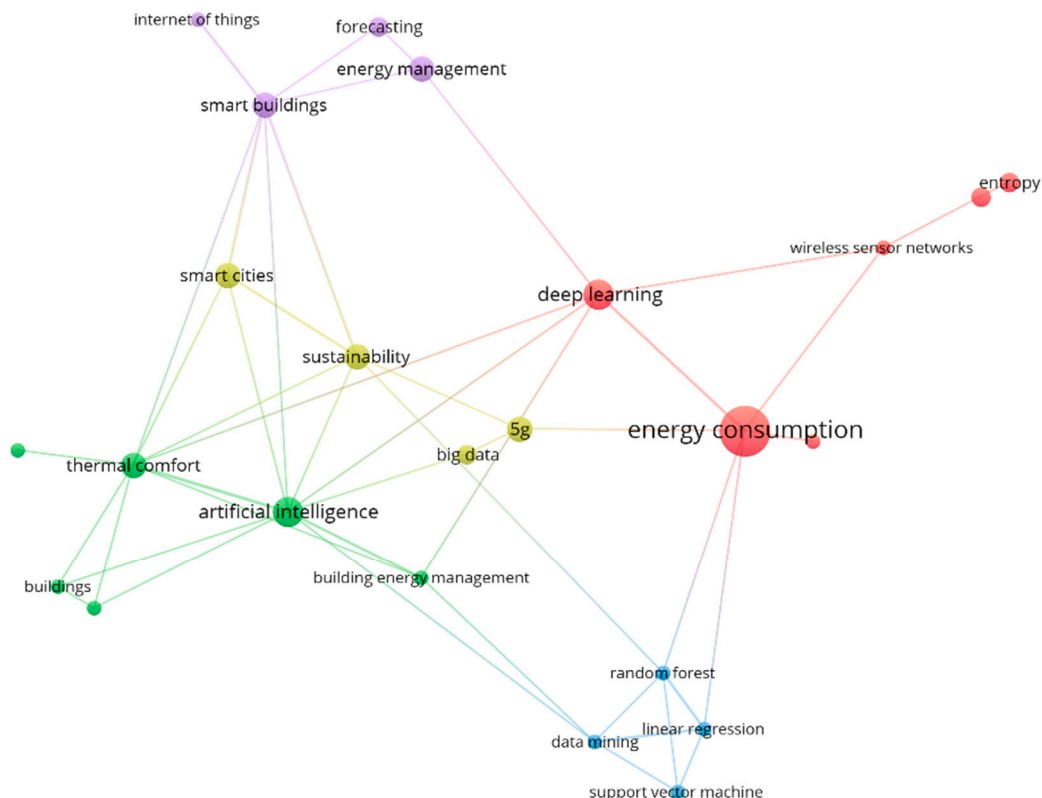

**Figure 7.** Keyword network. Source: the authors, based on data from Scopus.

In this search for sustainable energy consumption, the research focused on providing solutions using AI and ML in smart buildings [15], contributing to the sustainability of smart cities [63]. Therefore, researchers also aimed to develop smart devices such as an IoT for managing data on energy consumption towards reducing their consumption [24]. In this context, keywords such as 5G and wireless networkswe prevalent. The use of big data can provide information for designing predictive models for energy use in networks and thus help to build smart cities [64]. Other key terms such as RF, data mining, linear regression, and support vector machine (SVM) refer to methods for developing predictive models [21,33,65].

Regarding the relationship between keywords, it can be observed that the main associativity of words was between the following terms: energy consumption, deep learning, wireless network sensors, and entropy. The abovementioned refers to the use and design of sensors configured for monitoring and the use of data that allow for low energy consumption from Deep learning techniques and keeping the entropy of the system in equilibrium [60]; for example, a recent investigation presented the development of a PUF based on the pre-selection of entropy sources that serve as a power indicator to randomly select the back-end entropy sources that allow for complying with ultra-low power consumption and strong resistance to hacker attacks through machine learning (ML) [66].

The second most relevant conglomerate is that of smart buildings, energy management, the Internet of Things, and forecasting. This set of words refers to the use of prediction techniques from deep learning for the use of IoT devices in intelligent constructions to improve energy management [11]; that is, so that through IoT, it is possible to improve the management of energy resources mainly in buildings [15].

The following web of words is made up of the terms 5G, big data, sustainability, and smart cities. This network formed because some research focused on the development of systems that use big data for the design of 5G networks in order to contribute to the construction of smart and sustainable cities, as they use big data for data analy-

sis [34], including data associated with the energy consumption of devices that allow for 5G communication, such as access points, routers, and controllers, among others [60].

The following network is made up of artificial intelligence, thermal comfort, building energy management, and constructions. This network formed due to the investigations that were carried out, where they focussed on the development of air conditioning systems for buildings, using different artificial intelligence techniques to maintain adequate temperature control inside the building automatically [10], ensuring they are turned on only when the internal temperature of the building requires it and, in this way, making the lowest energy consumption possible [52]. The last word network is made up of technical aspects, where it refers to the different methods and techniques used in machine learning and statistical models that allow for the feeding of these methods; among these terms are random forest, data mining, linear regression, and support vector machine [21,47,65].

Figure 8 shows that the most used keywords varied by year. In 2012, the leading term was workload prediction; research mainly focused on using data to predict energy consumption and to develop execution classification approaches in buildings through a network of sensing, learning, and prediction agents for energy efficiency [67,68]. Then, in 2016, ML became relevant again given the concern for the high consumption of energy resources; therefore, the development of web-based systems emerged as a keyword because research was focused on using data analytics and cloud computing for sustainable building energy efficiency [41], in addition to designing energy efficiency models based on reinforcement learning [69].

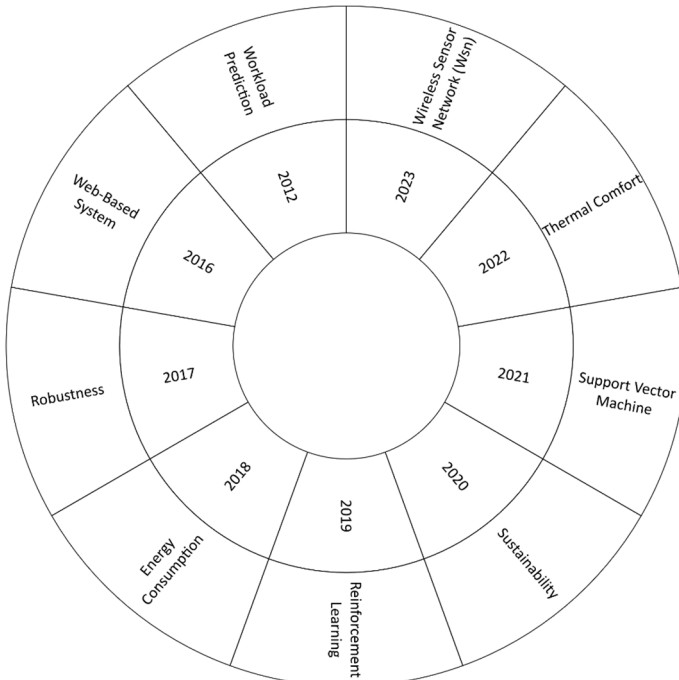

**Figure 8.** Keywords by year for energy efficiency and machine learning. Source: the authors.

In 2017, the term robustness emerged with the use and analysis of data for energy efficiency and robustness in datacentres, mobile devices, and networks [70]. In 2018, the term energy consumption emerged as energy efficiency research focused on ML techniques, simultaneously ensuring smartness, security, and energy efficiency in IoT sensors [71], and on data-driven optimisation of energy efficiency and comfort in an apartment [72]. In 2019, the most commonly used term was reinforcement learning, which refers to optimisation through deep reinforcement learning, for example, in energy efficiency systems [60].

In 2020, the most commonly used keyword was sustainability; several ML techniques were applied to design energy efficiency models for sustainable energy consumption [21]. Research was focused on improving energy efficiency in different contexts and systems,

such as irrigation [73], cooling [60], and building [7] systems, among others [71]. The term support vector machine emerged in 2021 as an ML approach for optimising the selection of anomaly detection functions in intrusion detection systems [47]. In 2022, the term thermal comfort emerged as the leading keyword in research on ML and DL methods for enhancing building energy efficiency and indoor environmental quality [10]. Research on energy efficiency was also conducted by implementing an AI model to predict room occupancy based on thermal comfort parameters [74]. Currently, the term wireless sensor network (WSn) is the keyword most commonly associated with the use of data on WSns for energy efficiency in these sensors [25].

The validity of different topics in research on energy efficiency and ML is shown in the validity quadrants of Figure 9. The term energy consumption is found in quadrant I. This term is associated with research on energy saving or responsible energy consumption to reduce the consumption and waste of resources or materials necessary to produce energy [27]. Research has also been conducted to improve the energy efficiency of carbon fibre manufacturing through waste heat recovery [59]. This term has become relevant given the importance of developing smart cities and their inclusion of Industry 4.0 technologies and sustainable development [9]. The terms AI and DL are found between quadrants I and II. AI has been used in different assisted techniques for managing control systems in buildings [52] or in wireless network energy efficiency [23]. DL is a deep learning technique applied to different systems for energy efficiency [75].

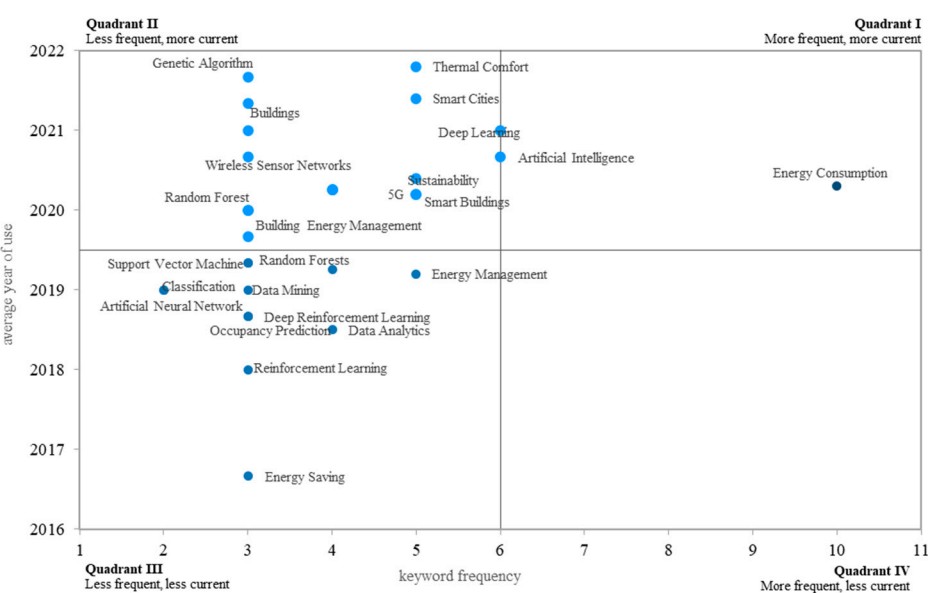

**Figure 9.** Keyword validity and frequency quadrants in the area of energy efficiency and machine learning. Source: the authors.

Quadrant I includes the most recent and frequent keywords, such as thermal comfort. One of the related studies developed a model based on thermal comfort parameters to predict room occupancy, so as to provide users with the best thermal environment and ensure energy efficiency [74]. The terms sustainability and smart cities and buildings are found in this quadrant. These three terms are associated with each other because they are used in research in which models are developed based on data mining for different ML applications. These techniques enable dynamic and automated development and resource optimisation [65]. The keywords buildings is commonly used in research on systems for improving energy use. Accordingly, some studies have focused on developing proposals for enabling buildings to adequately respond to changes in both indoor and outdoor environments [76].

Quadrant II encompasses the terms 5G and wireless sensor networks. The term 5G is applied in research on the use of ML for energy efficiency in mobile networks, which is one of the main challenges in developing next-generation networks based on the data provided by these networks [77]. In addition, 5G has been proposed as an alternative to reduce the consumption of energy needed to maintain the operation of high-performance computing networks [78]. The term genetic algorithms emerged as one of the techniques used to make energy-efficient programming decisions, as in the study by [79]. This model can be implemented in Matlab to assess makespan, execution time, resource utilisation, and energy quality and utilisation parameters to improve task scheduling and enhance sustainability.

Quadrant III includes less frequent and recent terms such as random forests, reinforcement learning and deep reinforcement learning, support vector machines, and artificial neural networks. These terms refer to ML techniques used for energy efficiency. In particular, DRL has been widely discussed in the analysis of data from wireless sensors, such as thermal comfort sensors, and in the optimisation of the energy efficiency of buildings [43,60,72].

The term data mining is related to the acquisition and use of data provided by different systems or equipment for predicting and designing energy efficiency models [41]. Data analysis has also been applied to predict the energy efficiency of air conditioning systems using deep neural network techniques [80]. The term occupancy prediction has been applied in research on occupancy prediction through ML for enhancing energy efficiency, air quality, and thermal comfort in the built environment [17]. The term energy saving has emerged in research on the prioritisation of large-scale residential energy efficiency enabled by ML [37]. Finally, quadrant IV, which includes the most frequent but least recent keywords, does not contain any elements, so decreasing concepts are not positioned.

Highlighting the validity of keywords recurrently used in this area of research and reflecting its thematic interest, Figure 10 shows the research agenda for energy efficiency and ML based on the use and relevance of concepts by year.

Research on topics such as energy saving, data analysis, and DRL as central topics of discussion are leaving the research agenda, falling behind in the discussion on energy efficiency, although the discussion on data analysis and DRL remains underexplored [72,81]. Terms such as occupancy prediction, data mining, and energy management, which were relevant between 2016 and 2018, are also leaving the research agenda [41,82]. These terms were used in research with different purposes, such as applying ML to develop a model for predicting the energy efficiency of seagoing ships transporting liquefied petroleum gas (LPG) [82].

Other terms, such as artificial neural networks, classification, machine support vectors, RF techniques, building energy management, and smart buildings, became relevant from 2018 to 2020; they are still widely discussed, but no longer valid. Research has been conducted on these themes, including on ambient analyses of energy efficiency in electrical smart grids [63] and on smart city planning by estimating the energy efficiency of buildings using extreme ML [83].

Some terms emerged from 2020 to 2021, i.e., linear regression, security, entropy, prognosis, sustainability, 5G, and AI. These terms are still widely discussed in research on 5G issues because studies have been mainly focused on energy efficiency data analysis and on 5G network quality and safety [84]. Other studies have been conducted to optimise communication over mobile networks, mainly energy efficiency optimisation and dynamic mode selection algorithms for D2D communication under HetNet in downlink reuse [85].

Lastly, the research agenda is focused on topics such as DL, as shown by the application of this technique in different energy efficiency models and in different contexts. As a case in point, quantum reinforcement learning has been used in energy-efficiency scenarios [86]. The research agenda also includes topics such as smart cities, thermal comfort, buildings, and genetic algorithms. The studies that have used these terms are related to temperature

regulation in indoor spaces, considering both environmental and occupancy factors to achieve energy efficiency in the built environment [17].

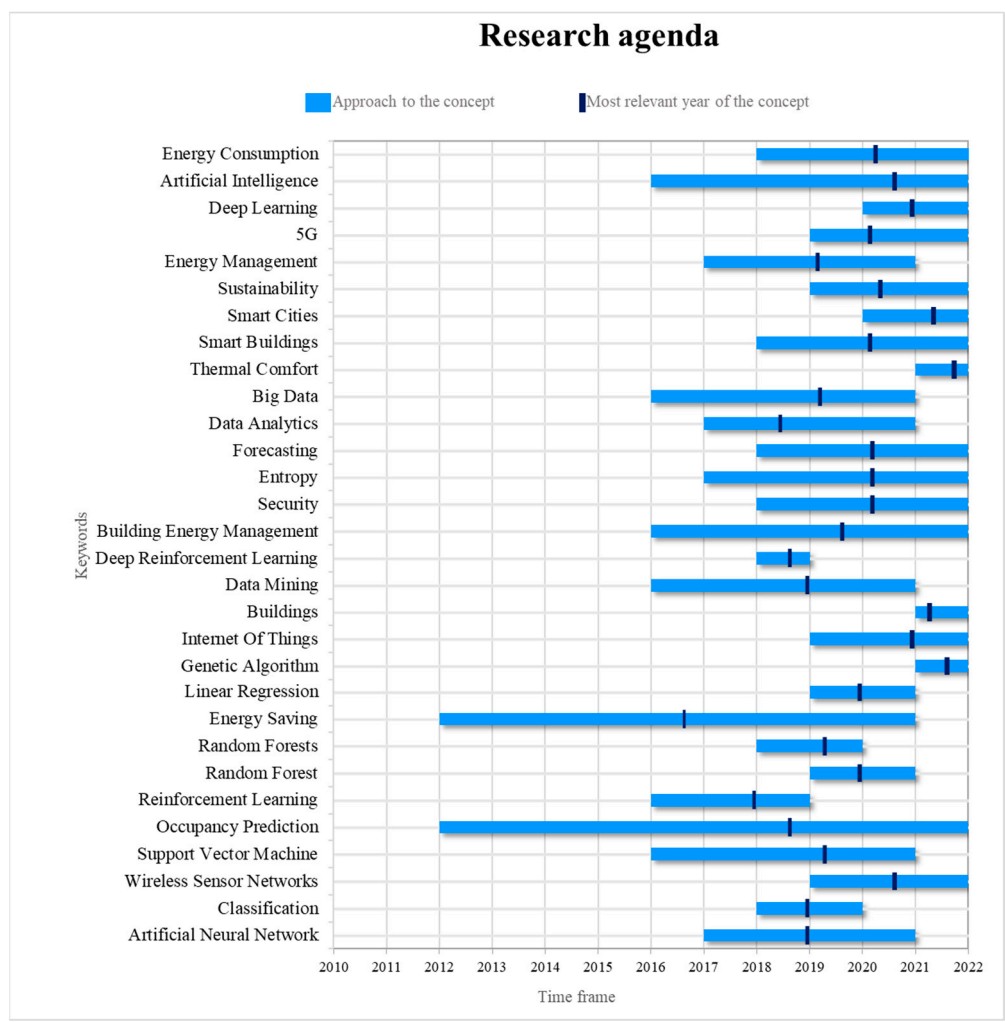

**Figure 10.** Research agenda in the area of energy efficiency and machine learning. Source: the authors.

Regarding the research agenda, for the periods between 2012 and 2016, the terms energy storage and occupancy prediction appeared; however, neither of these two terms had their greatest occurrence during this period. These two terms, however, have been two of the most used, the first refers to energy storage systems, either as a source of other systems or as a backup of other constant sources [32], while the second refers to research for the development of cooling or air conditioning systems according to the occupation of the buildings [17].

For the period between 2016 and 2021, the research agenda arose around terms such as artificial intelligence, big data, energy management of buildings, data mining, learning reinforcement, and support vector machines, which started in 2016. In 2017, energy management, data analytics, and artificial neural networks arose, and in 2018, investigative agenda makes its way into topics such as energy consumption, smart buildings, forecasting, security, deep reinforcement learning, random forests, and classification. In 2019, the agenda included the topic of linear regression, Internet of Things, sustainability, and 5G, and in 2020, the agenda also included the terms deep learning and smart cities.

For this period, which was the most active period in view of the emergence of research related to machine learning and energy efficiency, the most relevant terms due to their duration over time included artificial intelligence and building energy management, where

the former refers to the use of automation systems with the ability to generate intelligent responses for sustainable development, allowing for the adequate consumption and efficient management of energy, while the second refers to the proper distribution and use of energy in buildings.

On the other hand, the research agenda opened up starting from research oriented towards sustainable energy consumption, artificial intelligence, and web-based systems, towards the development of smart and sustainable cities and the use of more advanced programming techniques such as learning from deep reinforcement based on data analysis and thermal comfort, which aimed at not only the growth of the industry and society, but also the automation and optimization of equipment and systems that maintained user comfort. It is important to take into account that, as part of the development of smart cities, there is the Internet of Things specifically oriented towards the development of energy-efficient wireless sensors and the future of mobile networks [72], which also fall within the research agenda with the proposal of low energy consumption networks, but that provide a quality and secure service, where the data that are shared through them are not violated [76].

For the period from 2021 onwards, the research agenda makes its way with terms such as genetic algorithms, constructions, and thermal comfort, where this last term is the most recent in terms of occurrence in research. For these terms, genetic algorithms are based on development techniques based on the laws of behaviour and natural selection and genetics [85], which have been used in these investigations for automatic air conditioning in buildings [74] and in the use of energy according to the specific needs of devices and constructions [7,77].

Finally, considering the key terms, it is possible to classify the field of efficient energy and machine learning into three main categories that initially cover techniques associated with artificial intelligence and machine learning that have been mainly used in research on energy efficiency. The second category corresponds to the tools that have been used as a means for the development of devices or systems that seek energy efficiency. The third and last category corresponds to the purpose of the investigations, the application, or the purpose for which these investigations have been developed from the use of the techniques and tools used (Table 1). It presents the summary of the variables, as they are main trends according to the indicated categories, considering the results obtained for the agenda, network, square, and classification according to the year of the keywords.

**Table 1.** Keyword classification according to function.

| Techniques | Tools | Applications |
|---|---|---|
| Deep Learning | Sensors | Smart cities |
| Linear regression | Big data | Thermal comfort |
| Data mining | Data analytics | Energy management |
| Deep reinforcement learning | Wireless networks | Smart buildings |
| Genetic algorithm | Internet of Things | Sustainability |
| Artificial neural network | Forecasting | Energy consumption |
| Random forest | | Energy saving |
| Support vector machine | | |

Source. Own elaboration from the keywords.

## 5. Conclusions

The discussion on resource optimisation has prompted efforts to develop different methods for efficient resource use without reducing industrial productivity and without changing the routines and comfort of people. Accordingly, energy consumption was one of the first aspects that researchers addressed by searching for different alternative energy sources. However, thanks to advancements in information and communication technologies, research is now focused on not only the search for new energy sources, but also on the efficient use of current sources and resources.

Using different ML techniques enables not only the collection and analysis of data from equipment or tools that consume energy, but also data prediction and thus the development of different models for efficient energy based on other factors. The interest in applying ML techniques in the design and development of energy-efficient systems has increased considerably in the last 5 years due to the emergence of increasingly energy-demanding technologies such as smart buildings and cities and IoT. However, the development of smart cities and society demands sustainability; therefore, technological development is also used to promote sustainability, thereby enhancing energy efficiency.

Last, the research agenda is shifting from research geared towards sustainable energy consumption, AI, and web-based systems to the development of smart and sustainable cities, the use of more advanced DRL-based programming techniques, and thermal comfort, not only to grow the industry and society, but also to automate and optimise equipment and systems that maintain user comfort. In the development of smart cities, the IoT is specifically geared towards the development of energy-efficient wireless sensors and the future of mobile networks, which are also on the research agenda, with the proposal of low-energy-consumption networks that provide high-quality and safe services, where data are shared through them without violating personal data protection rights.

**Author Contributions:** Conceptualization, V.G.-P. and A.V.-A.; methodology, A.V.-A.; software, J.D.G.-R.; validation, R.B.G. and J.D.G.-R.; analysis, C.J.M.-V. and R.B.G.; investigation, V.G.-P., A.V.-A. and J.D.G.-R.; resources, V.G.-P. and A.V.-A.; data curation, C.J.M.-V.; writing—original draft preparation, V.G.-P.; writing—review and editing, A.V.-A.; visualization C.J.M.-V.; supervision, V.G.-P.; project administration, V.G.-P. and A.V.-A.; funding acquisition, A.V.-A. All authors have read and agreed to the published version of the manuscript.

**Funding:** This research was funded by Instituto Tecnológico Metropolitano (Colombia) and Universidad Señor de Sipán (Peru). The APC was funded by Universidad Señor de Sipán (Peru).

**Data Availability Statement:** The data may be provided free of charge to interested readers by requesting the correspondence author's email.

**Conflicts of Interest:** The authors declare no conflict of interest.

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
