# Peer review of "Machine-Learning Applications in Energy Efficiency: A Bibliometric Approach and Research Agenda"

_designs, 2023_

Round 1
Reviewer 1 Report
The paper is a bibliometric review studying trends in the machine learning-based design of electrical and electronic devices. The review covers 152 academic documents from Scopus and Web of Science databases. The introduction section defines the problem and sets the aims of the study, Materials and Methods section gives a comprehensive information on the methodology and data, the rest of the paper present clear results.
The paper is perfectly written and I do not see any reasons blocking its publication. If only, it would be interesting to evaluate the share of scientists involved in the use of ML for energy efficiency in the context of the country in order to evaluate interest of the scientific community (country) in these topics. Otherwise, no matter what topic, China and the USA will generate the most articles on it
Author Response
May 12, 2023
Dear
Design – Editorial Team
Kind regards
In accordance with the suggestions of the reviewers in our article “Machine Learning Applications in Energy Efficiency: A Bibliometric Approach and Research Agenda”, the following changes were made, properly marked with red letters in the article:
|
Reviewer |
Comment |
Response |
|
R2 |
Introduction: Include the relevant of Energy Efficiency in Sustainable development Goals (SDG 7) |
In relation to this aspect, a paragraph is added in the introduction that alludes to the contribution of energy efficiency to SDG 7 and SDG 11. |
|
R2 |
Materials and Methods: consider include an outline about the research methodology carried out. |
In relation to this aspect, it is mentioned that this scheme already exists in the article, and it is the flowchart suggested by the PRISMA-2020 declaration, where there is an article identification phase, where the sources of information and the number of results resulting from the search strategy (which is designed according to inclusion criteria), as well as the three phases of exclusion. |
|
R2 |
Figure 2: modify scale vertical axis (0% - 100%). |
Adjusts the scale of the vertical axis |
|
R2 |
Discussion: include Cluster analysis in the keyword network. |
The explanation about the associativity of the keywords is included in the final part of the cluster discussion, referring to each relevant subgroup. |
|
R2 |
Discussion: clarify the quadrants information in Figure 9. The information is incoherent. Quadrant II includes the most current keywords and less frequent. Quadrant IV includes the more frequent and less current keywords. |
Adjust the information in the quadrants, based on the graph and the reviewer's suggestion. |
|
R2 |
Discussion: Include different periods of research agenda: 2012-2016; 2016-2021, and 2021-present and explain the most important concepts. |
It is extended at the end of the discussion of the agenda, the reflection on the periods and the explanation of the most relevant key terms. |
|
R3 |
However, Iwould have expected the authors to follow the bibliometric analysis with at least one chapter coming a little kore in depth on the methodologies described in the 152 interest papers: if some common trends can be singled out, if some "stanhdard" methodology emerges or if the methodologies can be grouped in a limited number of different approaches, having each group important similarirties. Som in other words, I would have expected that the authors came a bit on the merit of the methodologies described in the interest papers, providing somer classification of them and clarifying the domain of utilization. If possible, I would ask the authors to add such extra chapter. |
At the end of the discussion, a section is added with the summary of the keywords according to the function of each one, classification in the categories of techniques, tools and application. |
|
R3 |
Finally, I notice that PRISMA is quoted different times in te paper, but I don't see neither an explanation of the (I imagine) acronym, nor a few words providing an overview of it. I ask the authors to add a brief description at the first occurrence of the PRISMA wording in the paper. |
The acronym PRISMA is defined in the abstract and again in the first line of materials and methods |
|
R4 |
1. The number of documents should be corrected in the abstract to 85 instead of 152. This inconsistency should be resolved throughout the paper to ensure accurate representation of the analysis. |
Considering figure 1, where the document selection process is shown and taking into account that the results of the Scopus and Web of Science databases were crossed, after filtering 152 results are obtained. This figure was unified throughout the document, taking into account the observation. |
|
R4 |
2. In Figure 2, the R-square value is not necessary for this type of diagram. The authors should consider removing it to avoid confusion and improve the clarity of the figure. |
The R-square of the figure is suppressed, according to suggestion |
|
R4 |
3. While the paper has the overall feel of a scientific work, it would be beneficial to provide a more systematic comparison and analysis of the 85 documents. It would be beneficial for the authors to delve deeper into the scientific results, methodologies, or applications presented in the analyzed documents. This could involve discussing the outcomes of specific studies, comparing the performance of various machine learning techniques in energy efficiency applications, or identifying common themes and patterns across the documents. By providing a more detailed analysis of the scientific results, the paper will offer a richer understanding of the state of research in machine learning applications for energy efficiency. |
Attending the suggestion, the analysis of the research agenda is broadened by periods in the discussion, including the cluster analysis of key terms. A summary of the key terms used is made, classifying them by categories according to their role in the field of energy efficiency. |
We look forward to your comments and hope to hear from you soon.
Thank you very much
_
The authors

Reviewer 2 Report
The article is very interesting and suitable for any reader in this field. The application of Machine Learning in Energy Efficiency improves the reduction of energy consumption. However, some recommendations should be considered for publication:
1. Introduction: Include the relevant of Energy Efficiency in Sustainable development Goals (SDG 7)
2. Materials and Methods: consider include an outline about the research methodology carried out.
3. Figure 2: modify scale vertical axis (0% - 100%).
4. Discussion: include Cluster analysis in the keyword network.
5. Discussion: clarify the quadrants information in Figure 9. The information is incoherent. Quadrant II includes the most current keywords and less frequent. Quadrant IV includes the more frequent and less current keywords.
6. Discussion: Include different periods of research agenda: 2012-2016; 2016-2021, and 2021-present and explain the most important concepts.
Author Response

(The authors gave the same response as above.)

Reviewer 3 Report
The paper presents a bibliometric analysis of paper published on the subject of Machine Learnming techniques applued to energy efficiency.
I have no remarks on the analysis methodology per se: standard bibliometric techniques were applied (and duly describe in the paper), which brough to isolate a nutshell of 152 papers of interest. However, Iwould have expected the authors to follow the bibliometric analysis with at least one chapter coming a little kore in depth on the methodologies described in the 152 interest papers: if some common trends can be singled out, if some "stanhdard" methodology emerges or if the methodologies can be grouped in a limited number of different approaches, having each group important similarirties. Som in other words, I would have expected that the authors came a bit on the merit of the methodologies described in the interest papers, providing somer classification of them and clarifying the domain of utilization. If possible, I would ask the authors to add such extra chapter.
Finally, I notice that PRISMA is quoted different times in te paper, but I don't see neither an explanation of the (I imagine) acronym, nor a few words providing an overview of it. I ask the authors to add a brief description at the first occurrence of the PRISMA wording in the paper.
Author Response

(The authors gave the same response as above.)

Reviewer 4 Report
The paper presents a bibliometric analysis of machine learning applications in energy efficiency, exploring research trends in the design of electrical and electronic devices. The authors have analyzed 85 academic documents from Scopus and Web of Science. The results highlight the growing interest in the subject since 2019, especially in the United States and China.
Notes for the Authors:
1. The number of documents should be corrected in the abstract to 85 instead of 152. This inconsistency should be resolved throughout the paper to ensure accurate representation of the analysis.
2. In Figure 2, the R-square value is not necessary for this type of diagram. The authors should consider removing it to avoid confusion and improve the clarity of the figure.
3. While the paper has the overall feel of a scientific work, it would be beneficial to provide a more systematic comparison and analysis of the 85 documents. It would be beneficial for the authors to delve deeper into the scientific results, methodologies, or applications presented in the analyzed documents. This could involve discussing the outcomes of specific studies, comparing the performance of various machine learning techniques in energy efficiency applications, or identifying common themes and patterns across the documents. By providing a more detailed analysis of the scientific results, the paper will offer a richer understanding of the state of research in machine learning applications for energy efficiency.
The paper may be published after minor revision.
Author Response

(The authors gave the same response as above.)
